# EFFICIENTLY ATTACKING MEMORIZATION SCORES

## ABSTRACT

Influence estimation tools—such as memorization scores—are widely used to understand model behavior, attribute training data, and inform dataset curation. However, recent applications in data valuation and responsible machine learning raise the question: can these scores themselves be adversarially manipulated? In this work, we present a systematic study of the feasibility of attacking memorization-based influence estimators. We characterize attacks for producing highly memorized samples as highly sensitive queries in the regime where a trained algorithm is accurate. Our attack (calculating the pseudoinverse of the input) is practical, requiring only black-box access to model outputs and incur modest computational overhead. We empirically validate our attack across a wide suite of image classification tasks, showing that even state-of-the-art proxies are vulnerable to targeted score manipulations. In addition, we provide a theoretical analysis of the stability of memorization scores under adversarial perturbations, revealing conditions under which influence estimates are inherently fragile. Our findings highlight critical vulnerabilities in influence-based attribution and suggest the need for robust defenses. All code can be found at https://anonymous.4open.science/r/MemAttack-5413/

## 1 INTRODUCTION

Online data market platforms, such as AWS Data Exchange (AWS), Dawex (Dawex), Xignite (XIgnite), WorldQuant (WorldQuant), are spaces where data is bought and sold. Concretely, there are three major entities in a data market: platforms, buyers, and sellers/providers (Kennedy et al., 2022; Alabi et al., 2025). The data market platform performs *data valuation* based on the acquired data from data sellers (Azcoitia & Laoutaris, 2022; Mehta et al., 2021; Agarwal et al., 2019; Fan et al., 2020; Wang & Jia, 2023). Sellers in data markets offer datasets (collections of information) that are valuable for businesses, researchers, or governments. Buyers look for data that can help them make better decisions, run more effective marketing campaigns, or improve products and services. *Data valuation* is released to buyers who use that information to buy data (Jung & Park, 2019; Ray et al., 2020).

Influence functions such as Shapley values are commonly used to price data (proportional to its valuation) in a data market (Yan & Procaccia, 2021; Song et al., 2019; Wang et al., 2020). A series of recent papers (Feldman & Zhang, 2020; Feldman, 2020; Brown et al., 2021) propose the *label memorization score* for supervised classification (in machine learning settings): in large datasets, a small subset of highly influential (memorized) training examples disproportionately affects the model's predictions and generalization capabilities, while the majority of examples have little to no impact. Clearly, this concept is relevant in data valuation, where the goal is to identify which training data points contribute most to a model's decision-making (i.e., samples with high memorization scores are more valuable). While the original proposal (Feldman & Zhang, 2020) is computationally expensive, various studies (Garg et al., 2023; Ravikumar et al., 2024; Jiang et al., 2020; Zhao & Triantafillou, 2024) propose efficient proxies.

Given the difficulty of obtaining high-quality data in data market platforms (and consequently their high value), there is a clear economic incentive for some sellers to manipulate the valuation scores (Alabi et al., 2025).

We aim to study how malicious data sellers can alter influence valuations in data markets. In particular, we look at the robustness of *memorization-based* approaches, by proposing multiple attacks based on distribution shift, stability notions, and decision boundary proximity.

Basu et al. (2021) previously study the robustness of influence functions, but do not provide theoretical guarantees and restrict their empirical analyses to neural networks.

Lai & Bayraktar (2020) provide an approach for adversarial robustness of general estimators (not influence estimators). Yadav et al. (2024) address the robustness of influence functions on adversaries that control the valuation algorithm, a strong assumption. In contrast to these works, we present the *first theoretical analysis* for vulnerability of memorized-based valuation functions. We particularly assert that when a trained algorithm exhibits high accuracy, we can characterize highly memorized samples with highly sensitivity. We take advantage of the inverse operation's naturally high sensitivity to motivate a simple, computationally efficient, and effective Pseudoinverse attack to produce highly memorized samples in image classification tasks. In comparison to prior work, we assume a much weaker adversarial model, one where the (malicious) data sellers only change the provided data, require no collusion, and are unaware of the exact (memorization-based) valuation algorithm. In our most effective attack, the adversary need not be aware of knowledge of the underlying data distribution! Figure 1 illustrates the threat model. The simplicity, yet effectiveness, of both our theory and attack demonstrate a fundamental vulnerability in data valuation for supervised learning.

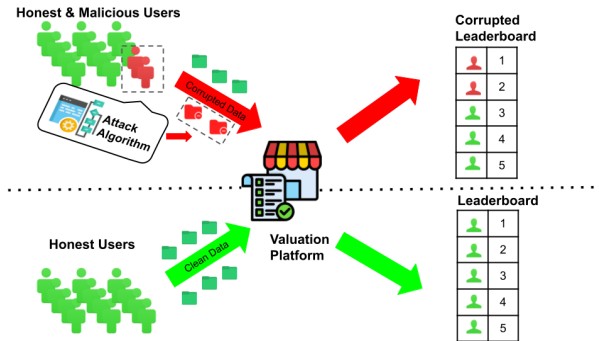

Figure 1: An overview of how data valuation functions can be attacked.

We summarize our contributions below:
- We develop a theoretical framework for analyzing vulnerability of memorization scores to adversarial manipulation and demonstrate its implications on potential data pricing (§ 4).
- We provide a simple and efficient Pseudoinverse attack that successfully alters memorization-based scoring of image classification tasks agnostic of underlying dataset and model architecture (§ 5).
- We verify our theoretical guarantees with experiments on image classification tasks across various convolutional network architectures on the MNIST, SVHN, and CIFAR-10 datasets (§ 6), and present additional support over more complex transformer architecture, higher resolution image datasets, and other data modalities in (§ C).

## 2 RELATED WORK

**Data Markets:** Azcoitia & Laoutaris (2022) give a survey on existing commercial data markets and their business models. Mehta et al. (2021) introduce pricing policies for enabling data economies. Agarwal et al. (2019) introduce a mathematical model for data marketplaces. Also, recent work surveys privacy and security vulnerabilities in data markets and offers possible design solutions (Alabi et al., 2025). Our work introduces a realistic threat model and calls for robust algorithmic solutions for future data markets.

**Data Valuation:** Shapley values are a classical concept in game theory which were first employed for the problem of data valuation in machine learning by Ghorbani & Zou (2019), termed Data Shapley Value, which aims to measure the influence of individual data samples on some specified performance metric. Due to computational overhead of computing Shapley Value, various influence functions have been proposed and adopted, including finding a core set of data examples (Yan & Procaccia, 2021), gradient-based approximations to the Shapley Value (Song et al., 2019), adaptations to federated learning setting (Wang et al., 2020), as well as non-Shapley based influence functions such as gradient tracing (Pruthi et al., 2020) and low-rank kernel approximation (Park et al., 2023). However, it is not

always clear what target for which the influence of data samples should be measured on. Common settings involve a specific test set on which the effects of influence are measured, but a definitive target test set might not be available or might change over time, or might not even be known in certain applications. Label memorization (Feldman & Zhang, 2020), the main metric of data valuation in this study, is a measure of self-influence that removes this particular challenge. Feldman (2020) shows that accuracy on training examples with label memorization scores is crucial to low generalization error, demonstrating that highly memorized samples strongly influence downstream tasks.

**Adversarial Attacks Against Memorization:** Privacy risk in the context of membership inference attack is closely related to memorization as shown previously by Choi et al. (2023). Carlini et al. (2022) find outliers to be more vulnerable to membership inference attack, whereas other prior work use poisoned images (Jagielski et al., 2020) or artificially crafted "canaries" for privacy auditing of language models (Carlini et al., 2019; Thakkar et al., 2020) to produce samples of high privacy risk, and subsequently high memorization scores. In comparison to prior adversarial studies, we present the first theoretical analysis characterizing highly memorized samples.

## 3 PRELIMINARIES AND NOTATION

### 3.1 QUERIES: SENSITIVITY AND ACCURACY

Let $\mathcal{Z}$ be a universe or domain. e.g., $\mathcal{Z} = \mathbb{R}^3$. Also, let $\mathbf{P}$ be a distribution over $\mathcal{Z}$ from which samples $\mathbf{z} = (z_1, \cdots, z_n) \in \mathcal{Z}^n$ can be drawn. We use the notation $z_1, \ldots, z_n \leftarrow_R \mathbf{P}$ to indicate that $z_1, \ldots, z_n$ is randomly drawn from $\mathbf{P}$. A mechanism, trained by the data evaluator or data platform, can answer *queries*, from some query family $Q$, about $\mathbf{P}$ or $\mathbf{z}$. For any $q \in Q$, we define the query answers on either the population level or sample-level:

$$q(\mathbf{P}) = \underset{\mathbf{z} \leftarrow_R \mathbf{P}}{\mathbb{E}} [q(\mathbf{z})] \qquad \text{and} \qquad q(\mathbf{z}) = \frac{1}{n} \sum_{i \in [n]} q(z_i).$$

Query families can be separated by their sensitivities, which quantifies how much the query changes when one or more elements of the input are changed. Let $\mathbf{z} \sim \mathbf{z}'$ denote that $\mathbf{z}, \mathbf{z}' \in \mathcal{Z}^n$ differ on at most one entry. For any query $q \in Q$ and neighboring $\mathbf{z} \sim \mathbf{z}'$, $\|q(\mathbf{z}) - q(\mathbf{z}')\|$ (the $\ell_2$-norm of the difference between $q(\mathbf{z})$ and $q(\mathbf{z}')$) is the sensitivity of the query $q : \mathcal{Z}^n \to \mathcal{Z}$. We can define $\Delta$-sensitive queries:

**Definition 1** ($\Delta$-Sensitive Queries). For $\Delta \geq 0$, $n \in \mathbb{N}$, these queries are specified by a function $q : \mathcal{Z}^n \to \mathcal{Z}$ where $\mathcal{Z} \subseteq \mathbb{R}^*$. Further, the queries satisfy

$$\|q(\mathbf{z}) - q(\mathbf{z}')\| \leq \Delta,$$

for every pair $\mathbf{z}, \mathbf{z}' \in \mathcal{Z}^n$ differing in only one entry.

$Q_\Delta$ is the set of all queries with sensitivity of at most $\Delta$.

As done in the literature on stability and adaptive data analysis (Bassily et al., 2016; Bousquet & Elisseeff, 2002; Shalev-Shwartz et al., 2010), $\mathcal{A}$ is a stateful algorithm with access to samples $z_1, \ldots, z_n \in \mathcal{Z}$. We can define an accuracy game between a stateful *adversary* $\mathcal{Q}$ and a mechanism $\mathcal{A}$, illustrated in Figure 2. The (opposing) goal of $\mathcal{Q}$ is to increase the error of the query answers provided by $\mathcal{A}$. Because $\mathcal{Q}$ and $\mathcal{A}$ are stateful, the queries and the query answers may depend on the history of past queries and past query answers.

A primary goal of machine learning is to train an algorithm/mechanism $\mathcal{A}$ that can accurately answer queries on $\mathbf{P}$. The training can use (independent) samples $z_1, \ldots, z_n \leftarrow_R \mathbf{P}$. Here, for each $i \in [n]$, $z_i = (x_i, y_i)$ corresponds to feature-label pairs, which are used to train $\mathcal{A}$. Then a query for $\mathcal{A}$ could be: *given* $\mathbf{z} \in \mathcal{Z}^n$, *what is the label* $y_{n+1}$ *for feature vector* $x_{n+1}$? Clearly, this can be encoded via the query function $q : \mathcal{Z}^n \to \mathcal{Z}$ where $q(\mathbf{z}) = q^*(\mathbf{z}, x_{n+1})$ and $q^* : \mathcal{Z}^n \times \mathcal{Z} \to \mathcal{Z}$ takes in samples and new feature vector $x_{n+1}$. The query answer is $z_{n+1} = (x_{n+1}, y_{n+1}) \in \mathcal{Z}$. For any $j \in [k]$, we measure query accuracy as $\|a_j - q_j(\mathbf{P})\|$ where $a_j$ is the answer by $\mathcal{A}$—based on the samples $\mathbf{z}$ and previous query answers—and $q_j(\mathbf{P})$ is the query answer on the population level. Note that the formalism goes beyond classification: we might want to estimate the mean of a population in which case the query is the population mean and the answer could be the mean of the samples $\mathbf{z} = (z_1, \ldots, z_n)$.

> Sample $z_1, \ldots, z_n \leftarrow_R \mathbf{P}$ and let $\mathbf{z} = (z_1, \ldots, z_n)$.
> For $j = 1, \ldots, k$
>   $\mathcal{Q}$ outputs a query $q_j \in Q$.
>   $\mathcal{A}(\mathbf{z}, \{q_t\}_{t=1}^j)$ outputs $a_j$.

Figure 2: The Accuracy Game $\mathsf{Acc}_{n,k,Q}[\mathcal{A}, \mathcal{Q}]$

Since queries are meant to return answers, we measure how accurate the answers are either on the population or specific samples.

**Definition 2** (Population Accuracy). *A mechanism $\mathcal{A}$ is $(\alpha, \beta)$-accurate with respect to the population $\mathbf{P}$ for $k$, potentially adaptively, chosen queries from $Q$ given $n$ samples in $\mathcal{Z}$ if for every adversary $\mathcal{Q}$,*

$$\mathbb{P}_{\mathsf{Acc}_{n,k,Q}[\mathcal{A},\mathcal{Q}]} \left[ \max_{j \in [k]} \|q_j(\mathbf{P}) - a_j\| \leq \alpha \right] \geq 1 - \beta.$$

In Definition 2, the randomness is over algorithms $\mathcal{A}, \mathcal{Q}$ and the distribution $\mathbf{P}$.

## 3.2 STABILITY NOTIONS

A variety of stability notions (such as Max-KL stability, KL stability, and TV stability) capture how sensitive an algorithm is to changes to its input (Hellström et al., 2025). In our work, we focus on Max-KL stability but our theoretical results can be ported to other notions of stability:

**Definition 3** (Max-KL Stability). *Let $\mathcal{A} : \mathcal{Z}^n \to \mathcal{R}$ be a randomized algorithm. We say that $\mathcal{A}$ is $(\epsilon, \delta)$-max-KL stable if for every pair of samples $\mathbf{z}, \mathbf{z}'$ that differ on exactly one element, and every $R \subseteq \mathcal{R}$,*

$$\mathbb{P}\left[\mathcal{A}(\mathbf{z}) \in R\right] \leq e^\epsilon \cdot \mathbb{P}\left[\mathcal{A}(\mathbf{z}') \in R\right] + \delta.$$

For any $\epsilon, \delta \geq 0$, it is easy to see that Max-KL stability is equivalent to $(\epsilon, \delta)$-*differential privacy* Dwork et al. (2006b;a).

Post-processing refers to the notion that a property (e.g., stability) is preserved after certain modifications. All the stability notions are preserved under post-processing:

**Lemma 1** (Post-Processing of Stability Notions (e.g., see (Bun & Steinke, 2016))). *Let $\mathcal{A} : \mathcal{Z}^n \to \mathcal{R}$ and $f : \mathcal{R} \to \mathcal{R}'$ be a pair of randomized algorithms. If $\mathcal{A}$ is {$\epsilon$-TV, $\epsilon$-KL, $(\epsilon, \delta)$-max-KL}-stable then the algorithm $f(\mathcal{A}(\mathbf{z}))$ is {$\epsilon$-TV, $\epsilon$-KL, $(\epsilon, \delta)$-max-KL}-stable.*

In addition, composition of disjoint databases preserves the stability notions:

**Lemma 2** (Post-Processing of Stability Notions (e.g., see (Bun & Steinke, 2016))). *Let $\mathcal{A}_1 : \mathcal{Z}^n \to \mathcal{R}$ and $\mathcal{A}_2 : \mathcal{Z}^n \to \mathcal{R}$ be a pair of randomized algorithms. If $\mathcal{A}_1, \mathcal{A}_2$ are {$\epsilon$-TV, $\epsilon$-KL, $(\epsilon, \delta)$-max-KL}-stable then $(\mathcal{A}_1(\mathbf{z}_1), \mathcal{A}_2(\mathbf{z}_2))$ is {$\epsilon$-TV, $\epsilon$-KL, $(\epsilon, \delta)$-max-KL}-stable if $\mathbf{z}_1, \mathbf{z}_2$ are disjoint.*

## 3.3 MEMORIZATION SCORES & PROXIES

Memorization reflects how much a model relies on memorizing specific training examples instead of generalizing.

**Definition 4** (Label Memorization (Feldman & Zhang, 2020)). *For training point $z_i = (x_i, y_i)$, the memorization score is:*

$$\mathsf{mem}(\mathcal{A}, \mathbf{z}, z_i) := \Pr_{h \leftarrow \mathcal{A}(\mathbf{z})}[h(x_i) = y_i] - \Pr_{h \leftarrow \mathcal{A}(\mathbf{z}^{\setminus z_i})}[h(x_i) = y_i].$$

Here, $h$ is the (randomized) classifier obtained from training algorithm $\mathcal{A}$ on dataset $\mathbf{z}$.

However, the first formulation of memorization proposed by Feldman and Zhang requires training thousands of models and is computationally prohibitive. Zhao & Triantafillou (2024) names two

computationally efficient proxies with empirically high correlation with memorization that we describe below to use in our experiments. We describe an additional proxy per risk scores computed from a membership inference attack given by Song & Mittal (2021).

**1. Input Loss Curvature-based Proxies:** Garg et al. (2023) observe that data samples with high loss curvature visually correspond to long-tailed, mislabeled or conflicting samples, which are more likely to be memorized.

**Definition 5** (Loss Curvature). The curvature of the loss function with respect to an input $x_i$ is:

$$\mathrm{curv}(\ell, i) := \mathrm{Tr}\left(\nabla_x^2 \ell(h_\theta(x_i), y_i)\right).$$

Implicitly, $\theta$ is a hypothesis trained from randomized algorithm $\mathcal{A}$, and $h_\theta$ is the model output from parameters $\theta$. $\ell$ is the loss function used to train models in $\mathcal{A}$, e.g. cross entropy.

**2. Learning Events-based Proxies:** The learning event proxies are first introduced by Jiang et al. (2020), a class of memorization score proxies designed to measure how quickly and reliably a model learns a specific example during training. The key intuition is that a specific example that is consistent with many others should be learned quickly as the gradient steps for all consistent examples should be well aligned. Zhao & Triantafillou (2024) find a strong link between these learning events proxies and memorization scores per the Spearman correlation coefficient. As done by prior work (Jiang et al., 2020; Zhao & Triantafillou, 2024), we aggregate the following cumulative statistics over each training epoch: confidence, max confidence, entropy and binary correctness. Full details of this class of attacks are described in Appendix B.

**Definition 6** (Learning Events Proxy). Given some per-sample event function $\phi(h_\theta(x), y)$ that depends on the learning hypothesis $\theta$ i.e. confidence, we define the cumulative training event proxy:

$$\mathrm{event}(\mathcal{A}, i, \phi) = \frac{1}{T} \sum_{t=1}^{T} \phi(h_{\theta_t}(x_i), y_i).$$

Here $\theta_t$ denotes the hypothesis learned at epoch $t$ in training algorithm $\mathcal{A}$.

**3. Membership Inference Attacks:** Membership Inference is a topic highly related with memorization. The goal of a membership inference attack is to recover whether a particular data entry was part of an unknown training set, either by using knowledge of the model, access to the model, or in a black-box setting. Choi et al. (2023) previously shows a theoretical link between Membership Inference advantage to memorization. Using a model trained on a certain dataset as the target model in an attack, we obtain risk scores for each data entry in the dataset—the probability of inclusion in the dataset—as a proxy for memorization scoring. The intuition comes from observing that highly memorized data entries are unlike representative data entries, and have higher probability of being identified by black-box model access. Song & Mittal (2021) give formulation for privacy risk score.

**Definition 7** (Privacy Risk Score). The privacy risk score $r$ of an input sample $z = (x, y)$ for the target machine learning model $h \leftarrow \mathcal{A}(\mathbf{z})$ is defined as the posterior probability that it is from the (random) set $\mathbf{z}$ after observing the target model's behavior over that sample denoted as $O(h, z)$, i.e.

$$r(z) = \Pr(z \in \mathbf{z} \mid O(h, z)).$$

These proxy scores are independent of each other, and each proxy operates on a different scale than the other, so the raw numbers are not meaningfully comparable. We give results comparing relative scoring in Appendix C.10.

## 4    MANIPULATING MEMORIZATION SCORES

We formally analyze the conditions under which data sellers can manipulate memorization scores. The contrapositive of Theorem 3 is that if the algorithm that is used to compute the memorization score is sufficiently accurate, then the memorization will be high for a family of queries. Our work builds on the theory of stability notions in the literature (Hellström et al., 2025; Bassily et al., 2016; Bousquet & Elisseeff, 2002; Shalev-Shwartz et al., 2010).

For our analysis, we note that *we do not control how the algorithm $\mathcal{A}$ is trained*! Thus, our adversarial model is natural and affords the algorithm designer the ability to respond to collusion by data sellers by modifying how $\mathcal{A}$ operates on the dataset.

> We show that if the classification algorithm $\mathcal{A}$ is very accurate on the population, then there *always* exist new examples from the data sellers that will lead to high memorization scores.

Let $q : \mathcal{Z}^n \to \mathcal{Z}$ be a query function. For any fixed dataset $\mathbf{z}$, the goal of the adversarial data seller is to consider if either $q(\mathbf{z}) \cup \mathbf{z}$ or $\mathbf{z}$ leads to a higher valuation score (via memorization).

For any algorithm $\mathcal{A}$, we study the following question: *what queries on the dataset would lead to high memorization scores*? In order to quantify the question, we consider the following memorization score on addition of a new example to the existing dataset:

$$\text{mem}(\mathcal{A}, \mathbf{z}, q(\mathbf{z})) := \Pr_{(x,y) \leftarrow q(\mathbf{z}), h \leftarrow \mathcal{A}(\mathbf{z} \cup q(\mathbf{z}))} [h(x) = y] - \Pr_{(x,y) \leftarrow q(\mathbf{z}), h \leftarrow \mathcal{A}(\mathbf{z})} [h(x) = y] \quad (1)$$

In Equation 1, what query functions $q : \mathcal{Z}^n \to \mathcal{Z}$ would lead to high memorization? We can measure sensitivity of a query as $\max_{\mathbf{z}, \mathbf{z}'} \|q(\mathbf{z}) - q(\mathbf{z}')\|$.

**Theorem 3** (See Theorem 4 in Appendix). *Let $Q_\Delta$ be a family of $\Delta$-sensitive queries on $\mathcal{Z}^n$. Fix $\delta \in [0, 1]$ and let the dataset size $n \in \mathbb{N}$ be such that there exists $\gamma > \delta$ such that $n \geq 1/\gamma$. Then for any $\alpha, \beta \in (0, 1/10)$, there exists algorithm $\mathcal{A}$ with memorization score (i.e., $\text{mem}(\mathcal{A}, \mathbf{z}, q(\mathbf{z})) \leq \delta$ from Equation 1) of at most $\delta$ such that $\mathcal{A}$ is $(\alpha, \beta)$-accurate for any query from $Q_\Delta$ but it must be the case that $\alpha \geq \gamma \Delta n$ and $\beta \geq \frac{\delta}{2\gamma}$.*

*That is, there exists algorithm with memorization score (Equation 1) of at most $\delta$ such that for $\mathbf{z} \leftarrow_R \mathbf{P}^n$ and query $q \in Q_\Delta$, $\Pr[\|q(\mathbf{z}) - q(\mathbf{P})\| \geq \gamma \Delta n] \geq \frac{\delta}{2\gamma}$, where $\mathbf{P}$ is a distribution over $\mathcal{Z}$.*

The inverse query $q(z) = z^{-1}$ has sensitivity that approaches $\infty$ (i.e., let the input be non-invertible or be 0 for one-dimensional input). This motivates a subset of our attacks (i.e., taking inverses of one or more examples). More generally, Theorem 3 implies that in order to avoid manipulation by data sellers, the algorithm $\mathcal{A}$ cannot be too accurate on the population level or the algorithm $\mathcal{A}$ must (formally) satisfy stability guarantees, such as max-KL stability.

**Full Details** For the full details of our proofs and analysis, see Appendix A.

## 5 EXPERIMENTAL SETUP

**Attacks:** We evaluate the robustness of data valuation methods under four distinct input-space attacks, each modifying input $x$ while preserving the label $y$. The attacks are presented from low to high fidelity i.e., how closely the adversarially modified image represents the original. Previous work (Xing et al., 2023) demonstrates that low-fidelity images can have high utility in synthetic data, which justifies the consideration of our visually aberrated attack images. Our attacks are motivated by distributional shift, stability notions and decision boundary proximity. A detailed description of how these attacks are implemented is given in Appendix B.

1. *No Attack:* Henceforth referred to as `None`, we use this label in our results section to denote unperturbed natural data samples.
2. *Out-of-Distribution Replacement Attack:* Henceforth referred to as `OOD`, this attack replaces in-distribution samples with out-of-distribution inputs from related datasets (e.g., FashionMNIST, CIFAR-10), testing sensitivity to semantic shifts. Intuitively, a small spurt of data drawn from a separate latent distribution will be long-tailed and memorized.
3. *Pseudoinverse Attack:* Henceforth referred to as `PINV`, this attack transforms input images by computing their Moore-Penrose pseudoinverses and normalizing to image-space ranges, inducing unnatural but structured distortions. Intuitively, by treating images as information signals for a model to parse, prior work (Brown et al., 2021) suggests that there are natural tasks for which any high-accuracy algorithm would need to memorize the majority of the samples. Motivated by our theoretical results, we take the pseudoinverse of an image matrix in order to result in memorization of the new image.
4. *Naive EMD Attack:* Henceforth referred to as `EMD`, this attack maximizes the Wasserstein distance between original and perturbed images using a greedy per-pixel binary search heuristic over RGB intensities. Intuitively, we seek to maximize the distance between the original and perturbed image, treating image data as probability distributions.

5. *DeepFool Perturbation Attack:* Henceforth referred to as DF, this attack applies the DeepFool algorithm (Moosavi-Dezfooli et al., 2016; Abroshan et al., 2024) to perturb each input minimally toward the classifier's decision boundary, forcing misclassification. Intuitively, we might expect a data point close to the decision boundary to be memorized by the model.

**Datasets:** We conduct experiments on three canonical image classification datasets: MNIST (LeCun et al., 1998), SVHN (Netzer et al., 2011), and CIFAR-10 (Krizhevsky et al., 2009). These datasets span grayscale handwritten digits (MNIST), real-world digit photographs (SVHN), and natural scene object categories (CIFAR-10), and thus collectively test attribution methods across increasing levels of input complexity and semantic variation. We also present supplemental results over ImageNet (Russakovsky et al., 2015) and AG News (Zhang et al., 2015) in Appendix C.12.

**Models:** We evaluate three standard deep neural net architectures: VGG-11 (Simonyan & Zisserman, 2014), ResNet-18 (He et al., 2016), and MobileNet-v2 (Sandler et al., 2018). These models represent different design paradigms: convolutional (VGG), residual (ResNet), and mobile-efficient (MobileNet). Each model is trained using standard data augmentation and optimization techniques (SGD with cross-entropy loss), with architectures matched to dataset resolution where applicable. We also present supplemental results over Vision Transformer (ViT) (Dosovitskiy et al., 2020) and BERT (Devlin et al., 2019) architecture in Appendix C.12.

**Experimental Setting:** We evaluate on the three proxies described in § 3.3. A detailed description of scoring implementation is found in Appendix C. Each attack is evaluated over $t = 5$ independent trials. In each trial, a randomly chosen subset, which we call the *attack set*, of data points from the base dataset is perturbed using the attack under consideration. We consider 3 sizes of attack sets: 10, 100, 1000. We then average the valuation scores of the perturbed samples and report the mean over trials as the final *attack score*.

**Hardware Used:** All experiments were conducted using NVIDIA A40 single GPU nodes. Training and attack procedures were implemented in PyTorch. More details can be found in Appendix C.

## 6 EXPERIMENTAL RESULTS

We aim to answer the following questions: (1) What attack is the most effective in manipulating memorization scores?; (2) Is this attack effective across dataset and model architecture settings?

As a quick summary, we empirically validate the following claims: (1) The high sensitivity queries produced by PINV perturb memorization scores (and proxies) to a greater extent than other attacks we consider; (2) PINV outperforms other non-motivated attacks across datasets and model architectures, shown through extensive experimentation over the MNIST, SVHN and CIFAR-10 datasets using convolutional network architectures and limited results over higher resolution datasets and textual data over ImageNet and AG News dataset using transformer architectures.

We also find that as the size of the attack set reaches $10^4$, PINV appears to lose advantage over OOD and EMD. However PINV does not require knowledge of the underlying data distribution (in comparison to OOD) and remains more computationally efficient (in comparison to EMD). While one might wonder if such adversarial modifications degrade model generalization, we find that the test accuracy does not significantly decrease; full testing analysis is presented in Appendix C.

Memorization scoring and its proxies are inherently metrics which have dependence on both individual data samples and underlying dataset. To compare performance the performance of different attacks, we present our results in accordance to *expected attack advantage* (EAA), our defined metric for comparing the memorization scoring perturbation capabilities for the attacks considered in our study.

In particular, EAA captures the expected improvement of memorization scoring between an adversary's attack in comparison to their honest uncorrupted (no attack) data as a baseline, motivated by our threat model of an adversary without knowledge of underlying data distribution. *Concretely, EAA captures the expected difference between scoring of data samples that are attacked and data samples that are produced by* None.

**Note:** Appendix C contains more visualizations and results over different experimental settings, including higher resolution datasets, textual data and transformer architectures.

## 6.1 MEASURING ATTACK EFFECTIVENESS

| Attack | Loss Curvature | | | Confidence Event | | | Privacy Score | | |
|---|---|---|---|---|---|---|---|---|---|
| | MNIST | SVHN | CIFAR-10 | MNIST | SVHN | CIFAR-10 | MNIST | SVHN | CIFAR-10 |
| None | 0.00±0.00 | 0.01±0.00 | 0.09±0.00 | 0.01±0.00 | 0.06±0.00 | 0.22±0.00 | 0.45±0.00 | 0.49±0.00 | 0.18±0.00 |
| OOD | 0.15±0.00 | 0.02±0.00 | 0.20±0.00 | 0.51±0.00 | 0.48±0.00 | 0.58±0.01 | 0.06±0.01 | -0.12±0.00 | 0.10±0.00 |
| PINV | **0.20±0.00** | **0.35±0.00** | **0.25±0.00** | **0.67±0.01** | **0.79±0.00** | **0.64±0.00** | **0.30±0.02** | **0.46±0.00** | **0.58±0.01** |
| EMD | 0.07±0.00 | 0.00±0.00 | -0.05±0.00 | 0.33±0.01 | 0.59±0.00 | 0.39±0.00 | -0.08±0.02 | -0.05±0.00 | -0.05±0.00 |
| DF | 0.00±0.00 | 0.01±0.00 | 0.00±0.00 | -0.01±0.00 | 0.01±0.00 | -0.03±0.00 | -0.02±0.01 | -0.03±0.00 | -0.02±0.00 |

Table 1: **EAA on ResNet-18 architecture with attack set size of 10.** PINV outperforms all other attacks by a significant margin across dataset and proxy.

| Attack | Loss Curvature | | | Confidence Event | | | Privacy Score | | |
|---|---|---|---|---|---|---|---|---|---|
| | MNIST | SVHN | CIFAR-10 | MNIST | SVHN | CIFAR-10 | MNIST | SVHN | CIFAR-10 |
| None | 0.00±0.00 | 0.01±0.00 | 0.09±0.00 | 0.01±0.00 | 0.06±0.00 | 0.23±0.00 | 0.47±0.00 | 0.49±0.00 | 0.19±0.00 |
| OOD | 0.13±0.00 | 0.02±0.00 | **0.14±0.00** | 0.62±0.00 | 0.52±0.00 | 0.61±0.00 | 0.08±0.00 | -0.10±0.00 | 0.09±0.00 |
| PINV | **0.14±.00** | **0.14±0.00** | 0.08±0.00 | **0.85±0.00** | **0.81±0.00** | **0.66±0.00** | **0.29±0.01** | **0.51±0.00** | **0.79±0.00** |
| EMD | 0.06±0.00 | 0.00±0.00 | -0.05±0.00 | 0.51±0.00 | 0.68±0.00 | 0.54±0.00 | -0.03±0.00 | -0.03±0.00 | 0.01±0.00 |
| DF | 0.00±0.00 | 0.00±0.00 | 0.01±0.00 | 0.00±0.00 | 0.00±0.00 | 0.00±0.00 | -0.02±0.00 | -0.01±0.00 | -0.02±0.00 |

Table 2: **EAA on ResNet-18 architecture with attack set of size 100.** PINV outperforms almost all other attacks by a significant margin across dataset and proxy.

Across our experimental settings, PINV outperforms the other attacks and produces samples that score highly relative to the base (unmodified) dataset. Tables 1 & 2 record the effectiveness of our four attacks on ResNet-18 models across difference choices of datasets and memorization scores and proxies (for an attack set sizes of 10 and 100). One can see the clear advantage of our theoretically motivated attack PINV in comparison to the other unmotivated attacks considered.

**An Explanation:** Our theoretical work suggests taking the inverse of an in-distribution sample strongly distinguishes it from the rest of the underlying dataset. We assert that it is the high sensitivity that causes the strongest scoring increase; additional experiments in Appendix C favorably compare PINV to random noise. On the other hand, while DF produces samples that are not meaningfully distinguishable from the underlying dataset, the attack is largely ineffective in perturbing memorization scoring. Based on Definition 4, one might believe that points that are most likely to change the decision boundary (i.e., those that lie close to it) would be more memorized. Thus, we also consider attack sets comprising of points that lie close to the decision boundary. Results of the effectiveness of DF across different attack sets is presented in Appendix C. We find that boundary starting points marginally improves the performance of DF, but not enough to beat out PINV which does not assume any prior knowledge of the underlying dataset. In line with our expectations, OOD performs well relative to baseline scoring. However, its performance compared with our PINV suggests that there is an underlying informational component to memorization that is not fully captured by distributional shifts.

## 6.2 LABEL MEMORIZATION

For completeness, we also provide a limited set of experiments for the effectiveness of the OOD and PINV attack on the more computationally intensive label memorization scoring per Feldman & Zhang (2020). Unlike Feldman & Zhang, we only use $n = 100$ number of models trained per scoring run due to computational constraints. Each model is trained using the same process as described in Appendix C. We demonstrate that the high sensitivity queries produced by the PINV attack raise label memorization scoring in accordance with our theoretical claims.

| Attack | MNIST | | | SVHN | | | CIFAR10 | | |
|---|---|---|---|---|---|---|---|---|---|
| Attack set size | 10 | 100 | 1000 | 10 | 100 | 1000 | 10 | 100 | 1000 |
| None | 0.01±0.00 | 0.01±0.00 | 0.01±0.00 | 0.08±0.00 | 0.08±0.00 | 0.08±0.00 | 0.23±0.00 | 0.23±0.00 | 0.23±0.00 |
| OOD | 0.08±0.00 | 0.02±0.00 | 0.01±0.00 | 0.04±0.00 | 0.02±0.00 | 0.01±0.00 | 0.02±0.00 | 0.02±0.00 | 0.01±0.00 |
| PINV | **0.62±0.00** | **0.36±0.00** | **0.20±0.00** | **0.58±0.00** | **0.45±0.00** | **0.13±0.00** | **0.43±0.01** | **0.23±0.00** | **0.09±0.00** |

Table 3: **EAA for label memorization across dataset over VGG-11 architecture.** PINV raises label memorization score significantly over base scoring and OOD attack.

| Attack | MNIST | | | SVHN | | | CIFAR10 | | |
|---|---|---|---|---|---|---|---|---|---|
| Attack set size | 10 | 100 | 1000 | 10 | 100 | 1000 | 10 | 100 | 1000 |
| None | 0.01±0.00 | 0.01±0.00 | 0.01±0.00 | 0.07±0.00 | 0.07±0.00 | 0.07±0.00 | 0.10±0.00 | 0.10±0.00 | 0.11±0.00 |
| OOD | 0.06±0.00 | 0.02±0.00 | 0.01±0.00 | 0.06±0.00 | 0.01±0.00 | 0.01±0.00 | 0.03±0.00 | 0.02±0.00 | 0.01±0.00 |
| PINV | **0.51±0.00** | **0.36±0.00** | **0.17±0.00** | **0.47±0.00** | **0.30±0.00** | **0.04±0.00** | **0.36±0.01** | **0.20±0.00** | **0.11±0.00** |

Table 4: **EAA for label memorization across dataset over ResNet-18 architecture.** PINV raises label memorization score significantly over base scoring and OOD attack.

| Attack | MNIST | | | SVHN | | | CIFAR10 | | |
|---|---|---|---|---|---|---|---|---|---|
| Attack set size | 10 | 100 | 1000 | 10 | 100 | 1000 | 10 | 100 | 1000 |
| None | 0.01±0.00 | 0.01±0.00 | 0.01±0.00 | 0.06±0.00 | 0.06±0.00 | 0.06±0.00 | 0.16±0.00 | 0.16±0.00 | 0.16±0.00 |
| OOD | 0.04±0.00 | 0.01±0.00 | 0.01±0.00 | 0.03±0.00 | 0.02±0.00 | 0.01±0.00 | 0.01±0.00 | 0.01±0.00 | 0.01±0.00 |
| PINV | **0.34±0.00** | **0.34±0.00** | **0.19±0.00** | **0.20±0.00** | **0.12±0.00** | **0.04±0.00** | **0.32±0.01** | **0.18±0.00** | **0.10±0.00** |

Table 5: **EAA for label memorization across dataset over MobileNet-v2 architecture.** PINV raises label memorization score significantly over base scoring and OOD attack.

### 6.3 SEMANTIC MEANING

It can be argued that images produced by the PINV attack have low semantic meaning and consequently will not be realistically considered in a data market setting after manual inspection. We respectfully believe that manual inspection alone provides limited insight into the actual contribution of a sample toward model performance and serves as a poor baseline for detecting adversarial samples. As shown in Wang et al. (2018), there exist many training samples that may appear semantically uninformative or even irrelevant to a human observer (i.e., easy to detect and discard) but nonetheless prove useful for generalization. Thus, it would be premature or misleading to discard or downweight such samples solely based on visual or semantic intuition.

Additionally, in many settings where data valuation is most critical e.g., active data curation, responsible dataset pruning, or cost-sensitive training, it is likely that the model's performance depends on the cumulative influence of a large number of individually subtle or redundant examples. This further reduces the practical viability of relying on human judgment alone. Manual inspection does not scale, especially when dealing with high-volume datasets or when the goal is to precisely quantify marginal utility of each point for fine-grained optimization.

### 6.4 TRANSFORMER ARCHITECTURE

While we focus on low resolution datasets and convolutional network architectures due to computational constraints of training models for our experiments, we assert that our main theory is agonistic of training algorithm and model architecture. We present a limited set of experiments over ImageNet data using ViT architectures in Appendix C.12 showing that PINV efficiently manipulates memorization scores for any training algorithm. We also present a small experiment over text classification dataset AG News over BERT transformer architecture that demonstrates the effectiveness of inverse-based attacks across other data modalities such as text in Appendix C.13.

## 7 CONCLUSION

In our work we present a theoretical justification for our framework of memorization score manipulation, as well as a suite of experimental results that demonstrate the vulnerability of memorization scores and its proxies to adversarial manipulation. We empirically find that a simple and efficient attack of taking a scaled Pseudoinverse of in-distribution image data is sufficient to successfully produce images with high memorization scores. We also give a theoretical starting point for further analysis of adversarial manipulation of memorization scores and other data attribution methods.

## ETHICS STATEMENT

Our study only uses publicly available data and does not incur privacy risk. We strongly feel that the results of our study will not cause any harm, and the technology presented poses no risk for misuse. LLMs were only employed for slight polish of prose; the main science is entirely contributed by the authors of this paper.

## REPRODUCIBILITY STATEMENT

Code and instructions to recreate the main experiments for our paper can be found at this link https://anonymous.4open.science/r/MemAttack-5413/. Our main theory is found in section 4 and completed in Appendix A. Details for experimental specifics and hyperparameters are found in Appendix C.

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
