# OpenReview forum: "Efficiently Attacking Memorization Scores"
_ICLR.cc/2026/Conference — Submitted to ICLR 2026_

### Official Review · Reviewer_jN5V · 2025-10-31

**Soundness:** 3
**Presentation:** 3
**Contribution:** 2
**Rating:** 6
**Confidence:** 3

**Summary:**

This manuscript examines whether memorization-based influence estimators
including direct label-memorization scores and several common proxies can be adversarially
manipulated under black-box access. The motivation is clear and timely: in emerging
data-valuation and data-market workflows, contributors have economic incentives to inflate their
samples’ “value,” and platforms increasingly rely on memorization-style scores to allocate
credit. The authors frame the problem through a sensitivity/stability lens, arguing that
sufficiently accurate models admit high-sensitivity queries that yield large memorization scores,
and they propose a simple input-space attack (a pseudoinverse, “PINV ,” transformation) that
requires only model outputs. Empirically, across MNIST, SVHN, and CIFAR-10 with standard
convolutional architectures (e.g., VGG/ResNet/MobileNet), PINV consistently increases
memorization scores and several proxy measures while reportedly preserving test accuracy. The
threat model is presented cleanly (Figure 1) and the experimental sweep is broad. Overall, the
paper is relevant, readable, and potentially impactful for the robustness of data-valuation
systems.

**Strengths:**

(i) A crisp and realistic threat model centered on black-box feasibility; (ii) a simple,
reproducible attack (PINV) that does not rely on gradients or training internals; (iii) broad
experiments across datasets and architectures, including both proxy metrics and
label-memorization proper; (iv) an attempt to aggregate results with an “expected attack
advantage” (EAA) notion; and (v) an accessible exposition that keeps the reader oriented to the
data-market use case.

**Weaknesses:**

The central theoretical development which is the
claim that sufficient accuracy implies the existence of high-sensitivity queries leading to high
memorization would benefit from more explicit notation (specific norms, probability spaces, and
quantifiers) and a self-contained statement/proof sketch in the main text. As written, the jump
from stability lemmas to a constructive black-box adversary is intuitive rather than formal; a
short proposition that instantiates the theory via the PINV transformation would close this gap.
On the empirical side, EAA is described narratively but lacks a precise definition and uncertainty
reporting in the main text; many table entries appear rounded to ±0.00, which obscures
variability and effect sizes. Because different proxies live on different scales, cross-proxy
comparisons are hard to interpret without normalization or per-proxy analysis; adding confidence
intervals/effect sizes and showing the correlation between proxy lifts and label-memorization
lifts would strengthen the claims. The paper asserts minimal impact on test accuracy, but a small
summary figure/table in the main text (rather than only in an appendix) would make this
immediately verifiable. Finally, the ethics statement feels overly categorical given that the paper
demonstrates a practical way to game valuation: please acknowledge plausible misuse and
outline mitigations (e.g., stability-aware scoring, robust estimators, adversarial screening before
awarding credit).

**Questions:**

The work sits at the intersection of adversarial ML and
data valuation. Prior studies have highlighted fragility in influence-style attributions and the
manipulability of certain valuation heuristics; this manuscript narrows the focus to
memorization-based estimators and argues that even black-box adversaries can induce
meaningful lifts. If the theoretical link between accuracy/stability and manipulable
high-memorization queries is clarified and tied more explicitly to the proposed attack, the paper
would offer a useful conceptual advance alongside its practical finding: platforms that depend on
memorization-style scores are at risk without additional defenses. For example, can the author comment on whether the influence estimation can be combined with prior defense method, such as "Trustworthy Machine Learning through Data-Specific Indistinguishability" and "Preventing Verbatim Memorization in Language Models Gives a False Sense of Privacy"?

---

> ### Author Response · Authors · 2025-11-17
> **Rebuttal by Authors**
>
> We would like to thank the reviewer for deciding to accept our work to the conference. We are glad our presentation of our simple attack and its theoretical backbone are appreciated. We would like to address the concerns you have in our response.
>
> 1. (**“The central theoretical…”**) We will edit the theoretical section for more clarity to address the concern raised by both this and another review.
> 2. (**“On the empirical side…”**) We will more precisely define EAA with mathematical notation as our main metric used in reporting our results as for a scoring mechanism $\texttt{score}$ and attack set $S\subseteq \mathcal{D}$, then we define
> $EAA:=\mathbb{E}\_{x\in S}[\texttt{score}(x)] - \mathbb{E}\_{x\in \mathcal{D}\setminus S}[\texttt{score}(x)]$.
> 3. (**“many table entries…”**) Although it may seem that we are obscuring variability in our tables, the truth is that when rounded to two digits of precision we have error bars of $\pm 0.00$. Since our tables are lacking empty horizontal space, we do not want to incur higher precision.
> 4. (**“Because different proxies live…”**) We acknowledge that comparing scores across proxies is not clear, but this is by design of the various proxy measures. The important result is the relative ordering of our attack PINV and the others remains the same across all settings and scoring mechanisms.
> 5. (**“The paper asserts minimal impact…”**) We will move the accuracy table to the main body accordingly.
> 6. (**“Finally, the ethics statement”**) We will edit our ethics statement to address this additional concern, we thank the reviewer for pointing out potential harm incurred by our study.
>
> We hope we have properly addressed your concerns, and we are happy to answer further questions.

---

> ### Author Response · Authors · 2025-11-20
> **Official Comment by Authors**
>
> We are still happy to answer any further questions. Thank you for submitting comments to help us improve our paper!

---

### Official Review · Reviewer_tdsN · 2025-10-31

**Soundness:** 2
**Presentation:** 1
**Contribution:** 2
**Rating:** 2
**Confidence:** 3

**Summary:**

The paper investigates the robustness of data attribution methods under adversarial manipulation. Specifically, it considers a threat model in which the data value is quantified by a sample’s memorization score, and the data provider seeks to construct examples that maximize this value, i.e. achieve higher memorization scores. To this end, the authors propose a theoretical framework for designing such attacks and develop a corresponding attack method based on the pseudoinverse of the original image matrix. They empirically evaluate how effectively the memorization score can be manipulated across three datasets and three variants of memorization metrics. Experimental results demonstrate that the proposed pseudoinverse-based attack, which aligns with their theoretical formulation, substantially outperforms existing heuristic baselines.

**Strengths:**

This paper studies a realistic and practical threat model where the adversary can play strategically to earn more benefits from the data company. The set-up is written clearly and easy to follow.

**Weaknesses:**

1. Connection between theory and the proposed algorithm (PINV). The connection between the theoretical result and the proposed algorithm PINV appears somewhat weak. In line 292, the paper notes that $q(z)=z^{-1}$ has sensitivity approaching infinity and, following the theorem, suggests that the pseudoinverse of an image (matrix) could serve as an effective attack method. However, if the underlying intuition is that high-sensitivity transformations lead to stronger attacks, then an element-wise inverse of each pixel value might seem an even more direct choice. The authors are encouraged to clarify why the pseudoinverse specifically, rather than other high-sensitivity transforms, is the theoretically motivated or empirically preferred option.
2. Clarity and soundness of Theorem 3. Theoretical Result (Theorem 3) raises several conceptual and clarity concerns.
    - Several statements in the theorem are difficult to interpret:
        - "let the dataset size $n\in\mathbb{N}$ be such that there exists $\gamma>\delta$ such that $n\geq 1/\gamma$": it is unclear what this condition is intended to ensure.
        - "There exists algorithm $\mathcal{A}$ with memorization score of at most $\delta$": the memorization score depends on both $\mathcal{A}$, $\mathbf{z}$ and $q(\mathbf{z})$. Does this statement mean that for some specific $\mathbf{z}$ and $q$, the score is bounded by $\delta$, or that bound holds universally for all $\mathbf{z}$ and $q$ under $\mathcal{A}$.
        - "$\mathcal{A}$ is $(\alpha, \beta)$-accurate ... but it must be the case that $\alpha\geq \gamma\Delta n$ and $\beta\geq \frac{\delta}{2\gamma}$" could perhaps be reformulated more cleanly, e.g. "$\mathcal{A}$ is $(\alpha,\frac{\delta}{2\gamma)$-accurate ...  $\forall\alpha\geq \gamma\Delta n and \alpha < 1/10$". Clarifying the intent of this statement would improve readability.
    - The case where $q$ s a constant function (with zero sensitivity, thus belonging to $\mathcal{Q}_{\delta}$ seems to contradict the theorem’s conclusion, as in this case $\mathrm{Pr}[|q(z)-q(P)|\geq \gamma\Delta n]=0$. The authors are encouraged to clarify how this case fits into the stated result -- it could be because I did not fully understand the results and the clarify of the theorem (e.g. the points illustrated above) might help.
3. Clarity and consistency of presentation. While the introduction and related work are clearly written, the later sections are difficult to follow due to inconsistent notation and missing definitions.
    - Inconsistent notation.
        - The function $q$ takes a dataset $z$ as input in most cases, but between lines 292–293, it appears to take an individual data point instead.
        - "$\mathcal{A}$ refers to the “model” in lines 154–158, but later denotes the “training algorithm,” for example in Theorem 3.
    - Missing details and unclear definitions.
        - The definition of the attack methods in Section 5 is not fully specified. For instance, for the main method PINV, how is the pseudoinverse defined for a three-dimensional image tensor (n_channel × width × height)?
        - Figure 2 defines the “Accuracy Game,” but this concept is not revisited later in the paper.
        - Section 3.2 introduces stability and provides two lemmas about its properties, yet these are not referenced elsewhere in the paper. The brief mention in the discussion of Theorem 3 is also unclear.
4. Practicality and detectability of the proposed attack. The proposed attack may be easily detectable, as the inversion of a natural image would likely fall far outside the natural image distribution. In the discussion section, the authors state that “manual inspection does not scale,” but it seems plausible that modern foundation models could readily distinguish natural images from their inverted counterparts. Additional discussion on this detectability issue and potential countermeasures would strengthen the practical relevance of the work.

**Questions:**

Please check the details in "Weaknesses"

---

> ### Author Response · Authors · 2025-11-17
> **Rebuttal by Authors**
>
> We want to thank the reviewer for their acknowledgement of our realistic threat model and problem statement. We hope to address the reviewer’s concerns below.
>
> 1. (**“Connection between the theory…”**) As pointed out, our theoretical result does not characterize the PINV attack as the best possible attack on memorization scores. We only use PINV as an example and acknowledge that other high sensitivity attacks may exist. The PINV attack is a computationally efficient attack that holds the desired high sensitivity property, which we will emphasize in our revision.
>
> 2. (**“Clarity and soundness of Theorem 3…”**) Thanks for pointing out where we can provide further clarifications of our theoretical work. We would also like to clarify the misconceptions pointed out. The main interpretation (which we can further clarify in the main body) of the parameter settings of the theorem is that: any sufficiently accurate algorithm inevitably admits high-sensitivity queries that attain high memorization gaps.
>
> - a. “Let dataset size $n\in\mathbb{N}$...”
> This is a precise condition in our theorem that ensures we have a large enough dataset so that the probability that the query on the sample and on the population differ (by the stated margin of $\gamma\Delta n$) is at least $\delta/(2\gamma)$. The proof of the theorem relies on partitioning the dataset into $\lceil 1/\gamma\rceil$ disjoint datasets/blocks each of size $\lceil \gamma n\rceil$. Clearly, we cannot partition the dataset into such blocks if $n < 1/\gamma$. We can discuss this technical point further in the main body. Thanks for pointing this out so that we may clarify.
>
> - b. “There exists algorithm $\mathcal{A}$ with memorization score of at most…”
> The proof is constructive: we construct an algorithm $\mathcal{A}$ with a memorization score of at most $\delta$. Please see appendix A for more details. We can further clarify our construction in the main body of our submission.
>
>
> - c. “$\mathcal{A}$ is $(\alpha,\beta)$-accurate…”
> This is exactly the $(\alpha, \beta)$ accuracy definition given in page 4 of our submission as Definition 2 (Population Accuracy). We will link the above statement to Definition 2 in our theorem statement with a Latex \ref command as well to make this clear.
>
> - d. In the case where $q$ is a constant function, $\mathcal{A}$ can not satisfy the condition of being $(\alpha,\beta)$-accurate, which does not contradict our main results.
> In particular, for any constant function $q$, the only distribution that it can be highly accurate on is a degenerate distribution. However, our definition 2 goes beyond such degenerate distributions. We can further clarify this point in the main body.
>
> 3. (**“Clarity and consistency of presentation…”**) We want to clarify some inconsistencies.
>
> - a. Inconsistent notation: In lines 292-293 the z is not bolded. We use a bolded z as a dataset consistently throughout the paper, but perhaps we can use a different symbol to clarify this distinction.
>
> - b. Missing details: The precise implementation attack is detailed in the appendix—specifically for a 3-channel image we take the pseudoinverse of each channel and stack them on top of each other. We can note this in the main paper for clarity;
> The accuracy game is used to define population accuracy which is used in our main theorem;
> We will add additional discussion of stability and how it relates to our main theoretical result and subsequent PINV attack in the revision. Concretely, we will make clear the theoretical connection between the literature of stability and the formulation of memorization by Feldman-Zhang.
>
> 4. (**“Practicality and detectability”**)  As we mentioned in our discussion section 6.3, manual inspection will not scale. We assert that human-judged “noise” can encode useful data for generalization as mentioned in the Wang et al. paper we cite, “Dataset Distillation”, so what may be “unnatural” to humans is not necessarily detectable by machines. There is also prior literature that documents failure of automated OOD detection, which suggests our attack might not be detected with high confidence [1]. We maintain our study presents a plausible threat vector, and nonetheless studies a successful manipulation of memorization scores which remains a novel contribution.
>
> We hope we have properly addressed your concerns, and we are happy to answer further questions.
>
> [1] https://arxiv.org/abs/2107.06908

---

> ### Author Response · Authors · 2025-11-20
> **Official Comment by Authors**
>
> We are still happy to answer any further questions. Thank you for submitting comments to help us improve our paper!

---

### Official Review · Reviewer_7HyW · 2025-11-01

**Soundness:** 2
**Presentation:** 1
**Contribution:** 2
**Rating:** 2
**Confidence:** 5

**Summary:**

The paper targets towards the problem of manipulating data valuation score, specifically, memorization score. It gives a high-level formal definition of the problem, and then provides a theoretical analysis on the existence of 'highly influential point' under constraints. The theorem mainly state that. for very precise algorithm the memorization score is not too high globally, it has a good probability that there exists a newly introduced sample that is of high memorization score. Built on this theorem, the paper proposes a method to take an (pseudo) inversion of an image as the input for higher memorization score. Results show that over several combination of model and dataset, this method shows good result on proxies of memorization score.

**Strengths:**

- The paper targets an important problem of attacking data valuation method
- The paper provides a theoretical analysis on the condition of model and input so that the input has a high chance to gain high memorization score.

**Weaknesses:**

- Firstly, the study of manipulating data valuation algorithms is not a brand-new idea. The paper itself has mentioned a related work [Influence-based Attributions can be Manipulated](https://arxiv.org/pdf/2409.05208) [1]. Moreover, there is an ICLR 2025 Paper that is not mentioned in this work, but the setup is very similar: [Adversarial Attacks on Data Attribution](https://arxiv.org/pdf/2409.05657) [2]. In fact, the threat model of this work is very similar to [2]. The only difference I see is that this work aims to attack memorization score, while [2] targets data valuation methods such as influence function and TRAK. The authors may need to justify their novelty beyond the theoretical analysis. Moreover, for a fairer comparison, the authors may need to consider adding [2] as a baseline attack method for comparison.

- Though there exists a theoretical analysis, it appears to me that the assumption might be too strong, that it may not apply to modern machine learning framework: that a model is precise enough, robust to a set of queries, and has moderate memorization scores globally. The authors need to demonstrate that for real ML system, it is at least close to this case, Otherwise, the importance of this theorem may be weakened because the assumption is too strong.

- The designed ``PINV`` method is flawed. Firstly, it is unclear how the theoretical analysis links with the ``PINV`` method. More importantly, the method, that is to take inversion of an image is very likely to output non-sense image, with a label that is non-sense to the image. For example the original input is a cat image with a label "cat", after inversion, the image becomes almost pure noise, but then the label "cat" is still there. This is NOT adding influential points, this is actually adding garbage input that is easy to be filtered by current machine learning system. In works such as [2], adding new input is done through perturbating image input with real semantic meanings by small amount of noise that is not distinguishable through human eyes, in other words the perturbation budget is very small. This also holds true for other baseline methods the paper choose in the experiment. However, the proposed method, which takes the inversion of the input, would have very large perturbation budget  $||x - x^{-1}||$. If one just want to add points that appear influential without caring about the budget or semantic meaning, there are countless ways to do it e,g, labeling a cat as dog/labelling noise that is generated by diffusion model with labels like cat or dog.

**Questions:**

See weaknesses,

Moreover,
- Could you provide some visualization example of images after ``PINV``?
- Could you list the budget required for baseline and your method?

---

> ### Author Response · Authors · 2025-11-17
> **Rebuttal by Authors**
>
> Thank you to the reviewer for their detailed write-up. We hope to address their concerns below.
>
> 1. (**“Firstly, the study of…”**) We reassert that our novel theoretical analysis analyzes memorization-based scoring functions, and the work proposed in [2] is aimed at influence functions. Regardless, we have provided experimental comparisons between our proposed PINV method and the Shadow and Outlier Attacks described in [2]. In each table “Advantage” refers to performance above the baseline of non-perturbed data points.
>
> Table 1: Loss Curvature Advantage on MNIST
>
> | Attack | Attack set size 10 | Attack set size 100 | Attack set size 1000 |
> |---|---|---|---|
> | PINV | 0.20 | 0.14 | 0.10 |
> | Shadow | 0.01 | 0.01 | 0.01 |
> | Outlier | 0.01 | 0.01 | 0.01 |
>
> Table 2: Confidence Event Advantage on MNIST
>
> | Attack | Attack set size 10 | Attack set size 100 | Attack set size 1000 |
> |---|---|---|---|
> | PINV | 0.67 | 0.85 | 0.70 |
> | Shadow | 0.24 | 0.22 | 0.19 |
> | Outlier | 0.21 | 0.19 | 0.19 |
>
> Table 3: Privacy Risk Advantage on MNIST
>
> | Attack | Attack set size 10 | Attack set size 100 | Attack set size 1000 |
> |---|---|---|---|
> | PINV | 0.30 | 0.30 | 0.20 |
> | Shadow | 0.05 | 0.03 | 0.02 |
> | Outlier | 0.04 | 0.01 | 0.01 |
>
> Using our memorization scoring functions, we demonstrate that our sensitivity-based attack method PINV outperforms the naive optimization-based attacks in [2]. Furthermore, the computational complexity of the PINV is minimal, whereas the Shadow Attack requires the training of multiple “shadow models” and many forward passes and the Outlier Attack requires many forward passes to attack an image. Finally, we argue that the PINV attack does not consider the threat model considered in [2], but actually a weaker threat model. In the Shadow Attack, knowledge of the training algorithm is required, and in the Outlier Attack query access to the central model for evaluating contribution is required. In our study, no knowledge of the input distribution nor training algorithm is required in our flagship PINV attack.
>
> 2. (**“Though there exists a theoretical…”**) The assumptions we consider are standard and we will edit our manuscript to clarify:
>
> - i. Data attribution is only relevant when the final model is precise enough: contributions to a poor model is not desirable for a central aggregator in the data market scenario;
>
> - ii. We do not assume the model is robust to any set of queries: our attack takes advantage of the fact that models might not be robust to high sensitivity queries such as the inverse function;
>
> - iii. In Feldman-Zhang’s seminal work introducing label memorization, they empirically demonstrate across settings for memorization scores to be mostly moderate with few highly scoring samples.
>
> 3. (**“The designed PINV method is flawed…”**) As mentioned in Section 4, our theoretical analysis concludes that given the moderate assumption of highly accurate classifiers, high sensitivity queries (for example the inverse function) will produce examples of high memorization. The Pseudoinverse is an “inverse-like” operation, so we consider such a simple attack for which we validate its success in further experiments. Finally, in our experiments we demonstrate that just as few as 10 examples in a 60,000 size training dataset such as MNIST can score extremely highly in memorization. Human inspection will not scale in the data market scenario where a large amount is bought and sold. As mentioned in Section 6.3, (Wang et al. 2018) demonstrate empirically that noisy data and/or data lacking semantic meaning can produce models with high accuracy. As such, the notion of a perturbation budget that captures human-defined semantic distance is flawed for automated detection of attacked data, since machine classifiers can use “garbage” data to learn “clean” data. As detailed in our Appendix C.11, the accuracy of our models trained on PINV-attacked data does not degrade by a significant margin, so our attack will not be easily detected by automated methods. Finally, other methods of generating “noise” that appears influential have not been empirically validated to our best knowledge, and the PINV method will be computationally cheaper than the mentioned diffusion forward pass. Additionally, our additional experiments beyond the main body demonstrate that PINV outperforms arbitrary uniform noise, which corroborates our theoretical analysis of sensitivity characterizing high memorization.
>
> We hope that we have properly addressed your concerns, and we are happy to answer any further questions.

---

> > ### Comment · Reviewer_7HyW · 2025-11-28
> > **Thanks for the response! and some follow up questions**
> >
> > Thanks for the response. My concern about second point is addressed.
> >
> > However, I have some further questions regarding point 1 and 3. Firstly, for point 1, could you demonstrate that memorization scores are utilized for data valuation in real-world cases?
> >
> > More crucially, for point 3, in fact there exists some automatic detection methods for such noisy input. For example, the cleanlab software (which is open-sourced) states that "(we can u)se cleanlab to automatically: detect data issues (outliers, duplicates, label errors, etc)", and I believe the noise with a label can simply be detected. The authors may want to demonstrate that their methods can pass such automatic detection.

---

> ### Author Response · Authors · 2025-11-20
> **Official Comment**
>
> We are still happy to answer any further questions. Thank you for submitting comments to help us improve our paper!

---

### Official Review · Reviewer_CvsY · 2025-11-03

**Soundness:** 2
**Presentation:** 3
**Contribution:** 2
**Rating:** 4
**Confidence:** 4

**Summary:**

The paper proposes attacks on memorization-based data valuation scores, claiming to make minimal assumptions.

**Strengths:**

(1) Section 3 is well-written.

(2) The authors do a good job of putting the paper together with theoretical motivations, followed up by experiments.

**Weaknesses:**

(1) The paper proposes implicitly to use memorization scores for data valuation, usually the literature relies on influence scores or SHAP. I am not convinced that memorization scores are the right metrics for valuation — if the training dataset only contains high-memorization samples will it be able to learn anything useful? Since high-memorization scores usually are outliers or mislabeled points.

(2) The crux of the paper is the PINV attack, but the images coming from this attack are unnatural. I don't think one has to target a particular image to create the pseudoinverse of or even create a pseudoinverse in practice -- one can minimally distort a tiny part of any image and this could perhaps be enough to memorize it.

My main reason for rejection is 1. If you could find another application for these attacks that would be better in my humble opinion.

**Questions:**

(1) The EMD attack is not clear. Aren't you trying to create perturbed images to begin with? so how is the distance calculated?

(2) The DF attack needs knowledge of the classifier. Are you treating this as a baseline or your own attack? Looks like the latter to me.

---

> ### Author Response · Authors · 2025-11-17
> **Rebuttal by Authors**
>
> Thank you for your review. We are happy to hear your acknowledgement of the clarity and presentation of our paper. We want to address the concerns raised in your review.
>
> 1. (**“The paper proposes implicitly…”**) We give an overview of other data attribution functions in the “Data Valuation” section of our “Related Work”, where we consolidate other literature on influence scores and Data Shapley value. To repeat our motivation for considering memorization scores, Feldman-Zhang label memorization is a notion of *self-influence* that doesn’t require a specified target for measuring influence or marginal contribution in Shapley values. Furthermore, in Feldman’s original work introducing label memorization, he theoretically links learning of highly memorized data samples with low generalization error; a key claim is that memorization is required for generalization contrary to popular intuition.
>
> 2. (**“The crux of the paper is…”**) We acknowledge the point that our theory doesn’t specify our PINV attack as being the only way to attack memorization scores. PINV over the entire image would have higher sensitivity than individually distorting a tiny part of an image, so our attack would still lead to higher memorization scores than the reviewer-proposed alternative. Regardless, we acknowledge that designing other highly sensitive queries (the key result characterizing high memorization scores in our paper) with additional natural constraints is a promising future direction, but beyond the scope of our current work of demonstrating vulnerability of memorization scores to adversarial attack.
>
> 3. (**“The EMD attack… The DF attack…”**) Because of space limitations in the main body, we detail the specifics of the EMD and DF attack in the appendix with pseudocode and further description. To repeat from the appendix, we maximize the Wasserstein or Earth Mover’s Distance from the original image, with the implicit understanding that the original image is representative of its class. The Deepfool attack is an attack we consider that requires additional knowledge of the classifier, based on the intuition of points laying on the decision boundary to be highly memorized. We use it to show that our theory of highly sensitive queries trumps “intuition” of memorization based on decision boundaries.
>
> We hope you are satisfied with our response and are happy to address any additional concerns.

---

> ### Author Response · Authors · 2025-11-20
> **Official Comment by Authors**
>
> We are still happy to answer any further questions. Thank you for submitting comments to help us improve our paper!

---

### Meta-Review · Area_Chair_BHm5 · 2026-01-01

**Summary:**

This paper studies adversarial manipulation of memorization-based scores, motivated by their potential use in data valuation and data markets. The authors present a theoretical perspective based on sensitivity and stability, and propose a simple black-box attack (PINV) that empirically inflates memorization scores across several datasets and models. The problem setting is timely, and the paper aims to highlight vulnerabilities in an increasingly relevant class of attribution metrics.

The main strengths of the paper lie in its clear motivation, well-defined threat model, and relatively extensive experimental evaluation. Several reviewers appreciate the breadth of experiments, the black-box nature of the attack, and the effort to connect memorization scores with broader questions of robustness and data attribution. One reviewer in particular finds the setup realistic and the empirical results consistent across architectures and proxies.

However, there are significant concerns that ultimately limit the impact of the work. A central issue is whether memorization scores are a meaningful or practically adopted metric for data valuation; the rebuttal does not convincingly establish real-world usage beyond theoretical connections to self-influence. The novelty relative to recent work on attacking data attribution methods is also limited, as the core idea of manipulating valuation scores via adversarial inputs is well explored. Additionally, multiple reviewers raise concerns about the weak and largely informal connection between the theoretical analysis and the specific PINV attack, as well as the clarity and soundness of the main theorem. From a practical standpoint, the proposed attack produces highly unnatural, noise-like images with large perturbations, which are likely to be detectable by existing automated data-cleaning or OOD-detection methods; the rebuttal does not provide strong evidence that such attacks would evade these defenses. Presentation and notation issues further hinder the evaluation of the theoretical contribution.

Overall, while the paper addresses an interesting and relevant question, the concerns regarding motivation, novelty, theoretical clarity, and practical realism are substantial. The rebuttal promises clarifications and revisions but does not sufficiently resolve the core issues raised by the reviewers.

**Reviewer Concerns:**

Several reviewer concerns were partially addressed in the rebuttal. The authors clarified parts of the theoretical assumptions and responded to questions about notation, definitions, and experimental setup, suggesting that some issues related to presentation and clarity could be improved with revisions. The rebuttal also included additional experimental comparisons to prior attack methods, which helps place the proposed approach in context from an empirical standpoint.

However, several important concerns remain unresolved. The rebuttal does not convincingly demonstrate that memorization scores are a practically meaningful or widely used metric for data valuation, leaving the core motivation insufficiently supported. Concerns about limited novelty relative to recent work on adversarial attacks against data attribution methods persist, as the primary distinction is the choice of target metric rather than a fundamentally new attack framework. The connection between the theoretical analysis and the specific PINV attack also remains weak, with the authors explicitly acknowledging that PINV is only an example rather than a theoretically derived or necessary construction. Finally, concerns about the practicality and detectability of the attack remain, as the rebuttal does not provide strong evidence that the highly unnatural, noise-like samples produced by the attack would evade existing automated data-cleaning or out-of-distribution detection methods.

**Reviewer Scores:**

Reviewer CvsY would likely keep their score unchanged, as their main concern about the relevance of memorization scores for data valuation was not fully resolved.

Reviewer 7HyW would likely maintain a reject score, since their concerns about novelty, practicality, and detectability of the attack largely remain after the rebuttal.

Reviewer tdsN would also likely keep their reject score, as the connection between theory and the proposed attack and the realism of the threat model were not convincingly addressed.

Reviewer jN5V might slightly lower their borderline accept score or remain neutral, given that several core issues raised by other reviewers remain unresolved and they explicitly indicated openness to rejection.

---

### Decision · Program_Chairs · 2026-01-26

Reject